# Recent Progress in Blue Thermally Activated Delayed Fluorescence Emitters and Their Applications in OLEDs: Beyond Pure Organic Molecules with Twist D-π-A Structures

**DOI:** 10.3390/mi13122150

**Published:** 2022-12-05

**Authors:** Yiting Gao, Siping Wu, Guogang Shan, Gang Cheng

**Affiliations:** 1State Key Laboratory of Synthetic Chemistry, Department of Chemistry, The University of Hong Kong, Hong Kong, China; 2Institute of Functional Material Chemistry and National & Local United Engineering Lab for Power Battery, Faculty of Chemistry, Northeast Normal University, Changchun 130024, China; 3HKU Shenzhen Institute of Research and Innovation, Shenzhen 518053, China

**Keywords:** TADF, MR-TADF, TSCT-TADF, metal-TADF, blue OLEDs

## Abstract

Thermally activated delayed fluorescence (TADF) materials, which can harvest all excitons and emit light without the use of noble metals, are an appealing class of functional materials emerging as next-generation organic electroluminescent materials. Triplet excitons can be upconverted to the singlet state with the aid of ambient thermal energy under the reverse inter-system crossing owing to the small singlet–triplet splitting energy (Δ*E*_ST_). This results from a specific molecular design consisting of minimal overlap between the highest occupied molecular orbital and the lowest unoccupied molecular orbital, due to the spatial separation of the electron-donating and electron-releasing part. When a well-designed device structure is applied, high-performance blue-emitting TADF organic light-emitting diodes can be realized with an appropriate molecular design. Unlike the previous literature that has reviewed general blue-emitting TADF materials, in this paper, we focus on materials other than pure organic molecules with twist D-π-A structures, including multi-resonance TADF, through-space charge transfer TADF, and metal-TADF materials. Cutting-edge molecules with extremely small and even negative Δ*E*_ST_ values are also introduced as candidates for next-generation TADF materials. In addition, OLED structures used to exploit the merits of the abovementioned TADF emitters are also described in this review.

## 1. Introduction

Organic light-emitting diodes (OLEDs) and the use of organic materials in light-emitting technology have caught the attention of researchers since Tang and VanSlyke’s pioneering work in 1987 [1] owing to their extraordinary merits, such as high efficiency, light weight, flexibility, and fast response [2,3,4,5,6]. A typical bottom-emitting OLED has a sandwich device structure, in which organic functional layer(s) are sandwiched between a transparent anode and a metal cathode. For high-efficiency OLEDs, the sandwiched functional layers usually consist of a hole-transporting layer (HTL), an emissive layer (EML), and an electron-transporting layer (ETL). In some cases, especially in phosphorescent and thermally activated delayed fluorescent (TADF) OLEDs, one or more auxiliary layers, such as a hole-blocking layer (HBL), a hole-injection layer (HIL), an electron-blocking layer (EBL), and an electron-injection layer (EIL), are added in the stack to further improve device performance. HBLs and HILs are used to confine carriers and excitons in the EML to avoid leaking current and the unwanted recombination of excitons outside the EML, and HIL and EIL are used to facilitate the injection of holes from the anode and electrons from the cathode to lower the driving voltage of the OLED [7]. In some cases, inorganic materials can be employed as HILs and/or EILs, such as MoO_3_, V_2_O_5_, LiF, Cs_2_CO_3_, etc. [8]. Such a sophisticated device structure is designed to fully exploit the potential of the emitting material, whose photoluminescent quantum yield (PLQY) and decay lifetime determine the performance of a well-designed OLED [9,10,11,12,13,14].

Light-emitting materials used in OLEDs have advanced rapidly in the last three decades, from fluorescent, phosphorescent, and triplet–triplet annihilation (TTA) materials to TADF materials [15]. Organic electroluminescence (EL) results from the radiative deactivation of excitons from the excited state to the ground state, which can occur with different spin multiplicities such as singlet or triplet excited states, resulting in fluorescence or phosphorescence. When OLEDs work under electrical excitation, singlet and triplet excitons are formed in the EML at a ratio of 1:3 [16], according to spin statistics. The quantum efficiency (QE) of an OLED, which refers to the numerical ratio of photons to injected electron-hole pairs [17], is used to assess its electricity-photon conversion efficiency, which can be further subdivided into internal QE (IQE) and external QE (EQE). IQE, the numerical ratio of total photons generated within the EML to the charge pairs injected, is directly determined by the emitting material [18]. In contrast, EQE is the numerical ratio of total photons emitting out of the device to the charge pairs injected. The chemical structure of the emitter and the associated functional materials can have significant impacts on the device efficiency, since photons are created within and emitted from the EML. IQE of fluorescent OLEDs is limited to 25% in traditional fluorescent emitters, which can only exploit singlet excitons for light emission. On the other hand, phosphorescent emitters that contain noble metals such as platinum, iridium, or osmium can harvest both singlet and triplet excitons through strong spin–orbit coupling, leading to high IQEs of up to 100% in phosphorescent OLEDs [19]. Nonetheless, further application of phosphorescent emitters is limited by the low abundance and high price of noble metals [20]. TTA fluorescent emitters can only achieve a maximum IQE of 62.5% by converting two triplet excitons into one singlet exciton, and their emission is mostly blue [21].

Alternatively, Yersin et al. [22] and Adachi et al. [23] developed inexpensive metal complexes and purely organic molecules, respectively, that are capable of harvesting both singlet and triplet excitons for emission, known as TADF. By reducing the overlap between the molecule’s highest occupied molecular orbital (HOMO) and the lowest unoccupied molecular orbital (LUMO), small singlet–triplet splitting (Δ*E*_ST_) can be achieved for efficient TADF. After singlet and triplet excitons are formed in the EML, the accumulated triplet excitons in T_1_ are transferred to S_1_ via a reverse inter-system crossing (RISC) process enhanced by thermal activation in the TADF mechanism [24]. By activating an effective up-conversion process of RISC from an excited triplet state to an excited singlet state, the TADF mechanism can also potentially achieve an IQE of 100% [25,26,27]. Compounds with suitable donor–acceptor building blocks are usually designed as TADF emitters, in which intramolecular charge transfer (ICT) with a small Δ*E*_ST_ is achieved by donor and acceptor moieties in the same molecule, allowing an effective RISC process to collect singlet emissions by converting triplets to singlets. In addition, twisted molecules, such as phenyl linkers between donors and acceptors, and a clear separation between the HOMO and LUMO distribution benefit TADF [7,9,28,29].

Along with judicious molecular designs, the performance of TADF-OLEDs has made significant progress. Remarkably, researchers have committed their attention to pure blue TADF emitters, which promise to be the final hurdle blocking the achievement of OLEDs for low-cost, highly efficient, and stable display illumination devices [30]. As one of the three primary colors, emitting blue is necessary for both display and lighting applications. However, neither fluorescent nor phosphorescent emitters are suitable candidates for blue OLEDs due to the low efficiency of the former and unsatisfactory stability of the latter. In this regards, TADF emitters could be a possible solution for low-cost, highly efficient, and stable blue OLEDs due to their potential high efficiency and stability. Early reviews have introduced the design concepts, the relationship between molecular structure and photophysical properties, and PL and EL performances of blue TADF materials, most of which are pure organic molecules with twist D-π-A structures [9]. However, new TADF emitters designed with novel concepts reported very recently have been sparsely covered in reviews. In this contribution, we focus on blue TADF materials possessing molecular structures other than pure organic with twist D-π-A structures. Recent advances of multiple MR-TADF materials are reviewed in Section 2, especially blue-emitting ones, that potentially address the issues of efficiency roll-off, quenching at high concentrations, and limited operational lifetimes that exist in early MR-TADF emitters. Newly emerged blue TADF emitters from through-space charge transfer (TSCT) processes are introduced in Section 3. A recent development in high-performance blue-emitting metal-TADF materials, especially those based on gold and copper, is briefly presented in Section 4. The outlook and perspective of next-generation TADF emitters with extremely small and even negative Δ*E*_ST_ are predicted in Section 5. In addition, OLED structures that are used to fully exploit the merits of the above-mentioned TADF emitters are also emphasized in this review.

## 2. Blue MR-TADF Emitters

Unlike the continuous enhancement in their efficiency, the color purity of blue TADF emitters has barely been improved due to their broad emission spectrum caused by a large Stokes shift, which is a result of the twisted conformation of the D-A molecules and the structural relaxation in the excited states. Alternatively, TADF can also be achieved in molecules with an MR effect, such as P- and B-doped rigid polycyclic aromatic hydrocarbons (PAHs) [31,32,33], from which narrowband emission with high color purity can be achieved due to the suppressed structural relaxation and lower vibronic coupling between the S_1_-S_0_ transition and the stretching/scissoring vibrations. Since the first MR-TADF material was reported by Hatakeyama and co-workers in 2016 [31], significant progress has been made in terms of efficiency and color purity [34]. Typically, OLEDs with the blue MR-TADF emitter ν-DABNA have demonstrated an emission maximum at 469 nm with an FWHM of 18 nm and an EQE_max_ of 34.4% [35]. In addition, owing to their small Stokes shift, MR-TADF materials have been widely used as terminal emitters in hyper-fluorescent OLEDs to improve the color purity of devices with phosphorescent or TADF emitters only [36,37,38,39,40]. Nonetheless, pronounced efficiency roll-off, severe quenching at high concentrations, and limited device stabilities remain formidable challenges to developing efficient blue MR-TADF emitters. In this section, we focus on the reports that are likely to address these issues.

The efficiency roll-off of OLEDs based on MR-TADF emitters is usually severe when compared to devices with conventional TADF emitters, due to the longer decay lifetime of the former. The long decay lifetime of MR-TADF emitters is mainly caused by their moderate Δ*E*_ST_ and RISC rate constant, which leads to an increased triplet-involved annihilation process [41,42,43,44]. Cao, Yang and colleagues presented a series of narrowband deep blue MR-TADF emitters (BN1-BN3, Figure 1) possessing a gradually enlarged ring-fused structure, which could simultaneously reduce Δ*E*_ST_, enhance *f*_osc_, and retain narrow linewidth of MR-TADF emitters [45,46]. By measuring the threshold energy of low-temperature fluorescence and phosphorescence spectra, the Δ*E*_ST_s of BN1, BN2, and BN3 were deduced to be 0.20, 0.16, and 0.15 eV, along with gradually accelerated RISC rates of 1.3, 2.6, and 25.5 × 10^4^ s^−1^ at room temperature. Together with the shorter lifetime (17.8 µs) of the delayed fluorescent and a higher PLQY of 0.98 in DBFPO thin film, BN3 displayed improved EL performance in hyper-fluorescent OLEDs with a structure of ITO/HAT-CN (5 nm)/TAPC (30 nm)/TCTA(15 nm)/mCBP (10 nm)/DBFPO: 25 wt% 3Cz2BN: 1 wt% BN1–BN3 (25 nm)/DBFPO (15 nm)/ANT-BIZ (30 nm)/Liq/Al. In these devices, BN1–BN3 were used as terminal emitters, while 3Cz2BN was used as a TADF sensitizer. The high EQE_max_ of 37.6% and EQE_1000_ of 26.2%, as well as the deep blue color with CIE coordinates of (0.14, 0.08) of the device with BN1, indicated that chromophore π-extension could decrease the decay lifetime of MR-TADF emitters without weakening its PLQY and broadening its FWHM, and may eventually address the efficiency roll-off issue.

For most MR-TADF molecules, the functional groups are mostly simple, rigid, and finitely extended to avoid spectral broadening. Such highly planar configurations may lead to intermolecular interaction-induced quenching, such as triplet–triplet annihilation (TTA) and triplet-polaron quenching (TPQ), at high dopant concentrations. Thus, extremely low dopant concentrations (usually less than 5 wt%) of MR-TADF emitters are used in OLEDs, which may lead to a relatively low maximum luminance and substantial efficiency roll-off at high luminance. To address this issue, host-featured segments have been introduced to form self-host MR-TADF emitters. In addition to averting intermolecular interactions, transporting abilities can also be readily tuned in such self-host emitters [44,47,48]. As shown in Figure 2, Xu and co-workers reported an ambipolar self-host featured MR-TADF emitter, tCBNDADPO, in which a B-N framework (tCBN) was substituted with an ambipolar A-D-A host segment (DADPO). The D-A type PO structures of DADPO significantly improved the TADF properties of tCBNDADPO, especially the accelerated singlet radiative rate constant of 2.11 × 10^8^ s^−1^, and exponentially reduced nonradiative rate constants, leading to a high PLQY of 99% for thin films with a high doping concentration of 30%. At the same time, narrowband blue emission with an FWHM of ≈ 28 nm was preserved in such film conditions. Taking advantage of the self-host feature, high EL performance of tCBNDADPO can be achieved in the OLED with a rather simple device structure of ITO/MoO_3_ (6 nm)/mCP (50 nm)/tCBNDADPO: DBFDPO (25 nm)/DBFDPO (40 nm)/LiF (1 nm)/Al (100 nm). A high EQE_max_ of 30.8% with coordinates of (0.14, 0.22) was achieved at a high concentration of 30%. Despite the efficiency roll-off (34.7% at a luminance of 300 cd m^−2^) still being unsatisfactory, the high concentration tolerance and the simplified device structure utilized may pave a path to the practical application of self-host MR-TADF emitters.

In addition to the serious efficiency roll-off, the relatively larger Δ*E*_ST_ and lower RISC process of MR-TADF emitters also make it difficult to control the triplet exciton density in the device, leading to a short device lifetime. To address this issue, Park and colleagues demonstrated that by the introduction of an additional blocking group to manage intermolecular interaction and the concentration quenching effect, a high efficiency and stable MR-TADF OLED is possible [49]. With an additional di-tert-butylphenyl (dtB) substituent on 2,12-di-tert-butyl-5,9-bis(4-(tert-butyl)phenyl)-5,9-dihydro-5,9-diaza-13b-boranaphtho[3,2,1-de]anthracene (t-DABNA) [50], 2,12-di-tert-butyl-5,9-bis(4-(tert-butyl)phenyl)-7-(3,5-di-tert-butylphenyl)-5,9-dihydro-5,9-diaza-13b-boranaphtho[3,2,1-de]anthracene (t-DABNA-dtB) was synthesized, as shown in Figure 3. With the increase in doping concentration in mCBP films, the PLQY of t-DABNA gradually decreased from 87% at 3 wt% to 54% at 10 wt%. At the same doping concentrations, the PLQYs of t-DABNA-dtB were 97 and 78%, respectively. The high PLQY of t-DABNA-dtB at high concentrations is attributable to its larger HOMO–LUMO overlap and suppressed concentration quenching, which is a result of the bulky substituent. By using t-DABNA or t-DABNA-dtB as the emitting dopant in OLEDs with an architecture of ITO /BCFN:HATCN (40 nm, 30 wt%)/BCFN (10 nm)/mCBP (10 nm)/mCBP:mCBP-CN:Emitter (30 nm, 3 wt%)/DBFTRz (5 nm)/ZADN (20 nm)/LiF (1.5 nm)/Al (200 nm), high EQE_max_ of over 25% were achieved. To further improve the device performance, especially the operational lifetime and efficiency roll-off, a sophisticated tandem device structure was adopted. High EQE_max_ of 30.1%, which slightly decreased to 28.8% at high luminance of 1000 cd m^−2^, was achieved in the tandem device with t-DABNA-dtB, together with a pure blue emission color with CIE coordinates of (0.116, 0.116) at 10,000 cd m^−2^. As shown in Figure 4, a long device lifetime, LT_95_, of 502 h at an initial luminance of 1000 cd m^−2^ was achieved in the tandem device. This lifetime is one of the best results for blue OLED lifetimes as reported in the literature, showing a bright future for blue MR-TADF in practical applications.

## 3. Blue TSCT-TADF Emitters

In addition to emitters in a twisted D-π-A structure via through-bond charge transfer (TBCT), TADF can also be the result of TSCT processes that usually take place between a pair of exciplex-forming D and A molecules, which is known as inter-TSCT. On the other hand, TSCT-TADF emitters (intra-TSCT) are molecules where the D and A segments are linked in a V or U shape, in which TSCT between D and A segments is possible. When compared with TBCT emitters, TSCT emitters have reduced electronic communication and coupling between D and A moieties, which benefits the elevation of the singlet and triplet levels and thus blue emission. At the same time, high PLQYs could easily be achieved in the TSCT molecules with intrinsic intramolecular noncovalent interactions [51,52]. The distance and orientation between D and A segments, determined by the linker, is a crucial factor in the excited state dynamics of a TSCT-TADF emitter [53]. Of the various types of alignment of D and A segments in TSCT molecules, face-to-face alignment with short distance is the most favorable for efficient TADF emission. However, only a few blue examples have been reported among high-performance OLEDs with TSCT-TADF emitters [54,55,56,57,58,59,60].

As shown in Figure 5, Wang and co-workers demonstrated a TADF polymer in which the TSCT effect takes place between pendant D and A units on a nonconjugated polyethylene backbone [59]. Unlike conventional conjugated D-A polymers, there is no direct conjugation between D and A units in the TSCT-TADF polymer because of its nonconjugated architecture, which is favorable for blue emission. Two polymers with the same A unit of 2,4,6-triphenyl-1,3,5-triazine (TRZ) but different D units of 9,9-dimethyl-10-phenyl-acridan (Ac) or 9,9-bis(1,3-ditert-butylphenyl)-10-phenyl-acridan (TBAc) were synthesized and compared. As a result of its steric 1,3-ditert-butylphenyl groups, the acridan unit is separated from the triazine unit in the TBAc-based polymer, while acridan approaches triazine in the Ac-based polymer, which determines the different D-A distance in different polymers. As expected, since TSCT is highly sensitive to the D-A distance, the distinct TSCT effect and TADF feature were only displayed in the Ac-based polymer, where a small Δ*E*_ST_ of 0.019 eV and high PLQY up to 60% in the film state were achieved. On the other hand, no TSCT effect and only fluorescence emission were found in the TBAc-based counterpart. When the polymer with 95 mol % Ac and 5 mol % TRZ (P-Ac95-TRZ05) was used as the EML in the device with a structure of ITO/PEDOT: PSS (40 nm)/P-Ac95-TRZ05 (40 nm)/TSPO1 (8 nm)/TmPyPB(42 nm)/LiF (1 nm)/Al (100 nm), blue emission with CIE coordinates of (0.176, 0.269) and an EQE_max_ of 12.1% were achieved, which is the first example of blue TSCT-TADF polymer for solution-processed OLEDs.

Zhang, Duan and colleagues adopted a xanthene bridge to construct space-confined face-to-face D-A alignment and minimize the D-A distance to 2.7–2.8 Å to strengthen the electronic interaction between weak D and A, which is required for efficient blue emission [39]. The targeted TSCT-TADF emitters are shown in Figure 6, exhibiting peaks around 460 nm, PLQYs > 90%, and a k_rs_ of nearly 107 s^−1^. The EL of such blue emitters was investigated in OLEDs with a structure of ITO/HAT-CN (5 nm)/TAPC (30 nm)/TCTA (10 nm)/mCP (10 nm)/PPF: 30 wt% emitters (24 nm)/PPF (10 nm)/BPhen (30 nm)/LiF (0.5 nm)/Al (150 nm). Blue emission with an EQE_max_ of 27.8% and CIE coordinate of (0.17, 0.29) were achieved when dCz-Xo-TRZ was used as the emitter. dCz-Xo-TRZ was further used as a sensitizer in hyper-fluorescent OLEDs with the well-known MR-TADF emitter, v-DABNA, as the terminal emitter. A high EQE_max_ of 34.7% with a CIE_y_ coordinate of 0.15 and FWHM of 19 nm was realized in the resulting device.

Despite the high efficiency achieved in blue TSCT-TADF emitters [39,55,57], their color purity is still far from meeting the ultrapure-blue requirements of CIE (0.14, 0.08) defined by the NTSC, as well as CIE (0.131, 0.046) for the standard of Rec. 2020 [60], due to the large Stokes shift and broad emission with large FWHM in conventional TADF emitters. Alternatively, as mentioned in Section 2, the FWHM can be effectively narrowed by using MR-TADF emitters due to their rigid polycyclic aromatic framework. Ren and co-workers designed TSCT-TADF emitters by combining acridine derivatives and electron-withdrawing boron/oxygen heterocycles [60]. As depicted in Figure 7, by controlling the intramolecular stacking of rigid heteroaromatic compounds, deep blue emission with a narrow FWHM could be achieved. By using AC-BO, QAC-BO, and Cz-BO as emitters in OLEDs with a structure of ITO/HAT-CN (20 nm)/TAPC (40 nm)/mCP (10 nm)/20% emitters: DPEPO (20 nm)/DPEPO (10 nm)/TmPyPB(50 nm)/LiF (1 nm)/Al (150 nm), the EL properties of AC-BO, QAC-BO, and Cz-BO were investigated. With decreasing stacking distance, emission maxima of AC-BO, QAC-BO, and Cz-BO were significantly blue-shifted from 456 to 412 nm, while the FWHM narrowed from 71 to 43 nm, leading to CIE coordinates of (0.147, 0.122), (0.145, 0.076), and (0.163, 0.034), respectively. The device with AC-BO shows an EQE_max_ of 19.3%, the best performance among all deep blue TSCT-TADF emitters so far [60]. The device with QAC-BO achieved an EQE_max_ of 15.8%, making it the first high-efficiency TSCT-TADF material to meet the ultrapure blue requirements of CIE (0.14, 0.08) defined by the NTSC. Nonetheless, the relatively low maximum luminance of less than 1000 cd m^−2^ and pronounced efficiency roll-off of more than 60% at 100 cd m^−2^ hinders the practical application of these deep blue emitters.

## 4. Blue Metal-TADF Emitters

Despite their rapid development, the long-term stability of OLEDs with organic TADF emitters is still inferior when compared with those with phosphorescent or fluorescent emitters, due to the long-lived triplet excitons of organic TADF molecules resulting from their inefficient spin–orbital coupling (SOC) [61,62]. Alternatively, by taking advantage of the strong SOC of phosphorescent organometallic complexes, metal-TADF emitters may achieve high emission quantum yields and fast radiative decay rates, and thereby eventually find practical applications in the OLED industry. Since Yersin and co-workers demonstrated the first TADF emission from a [(pop)Cu(N^N)] complex, in addition to Cu(I) emitters [63,64], second and third row transition metal complexes, such as those of Pd(II), Ag(I), Au(I), and Au(III), have been developed as TADF emitters [65,66]. Among the reported metal-TADF emitters, those based on two-coordinate carbene-metal-amide (CMA) emitters and tetradentate Au(III) complexes have displayed the best performance in terms of efficiency and stability [67,68]. Nonetheless, performances of blue-emitting OLEDs with metal-TADF were relatively inferior when compared to those of green, yellow, and red in terms of efficiency and color purity, until a recent publication presented blue-emitting metal-TADF emitters based on Cu(I) and Au(I) complexes [40,62,65,66,69,70,71,72].

As shown in Figure 8, by substituting CF_3_ and *^t^*Bu for the carbazole substitution of the CMA(Au) emitter they reported earlier [67], Romanov, Credgington and colleagues developed a highly efficient blue-emitting metal-TADF emitter Au(I)-1 [71]. Since the HOMO is almost entirely located on the Cz donor of the (^Ad^CAAC)AuCz archetype, the introduction of electron-withdrawing groups to this moiety influences the HOMO more than the LUMO. It widens the HOMO–LUMO gap, shifting it towards blue emission [67]. A high quantum yield of 0.96 with an emission maximum located at 495 nm was achieved in Au(I)-1 in toluene solution. Considering that the high polarity of CMA compounds makes their emission energies sensitive to their molecular environment, allowing the “tuning” of EL by suitable host media from green to sky blue, the emission maximum of Au(I)-1 can be tuned from 484 nm in neat film to 479 and 464 nm when doped in o-CBP and DPEPO hosts, respectively. The EL performance of Au(I)-1 was characterized by an OLED with a device structure of ITO/TAPC (40 nm)/o-CBP (5 nm)/EML (30 nm)/TSPO1 (40 nm)/LiF (1 nm)/Al (100 nm). In addition, since CMAs do not suffer from strong concentration quenching in the solid state, attributed to the lack of close metal–metal contacts, host-free OLEDs with neat Au(I)-1 as the EML have also been fabricated. Similar to the trend in the PL spectra, the device with the high-polarity DPEPO host shows the best color purity, with a maximum located at 450 nm, leading to CIE coordinates of (0.17, 0.17) along with a high EQE of up to 20.9%. Encouragingly, a high performance with an EQE of up to 17.3% and CIE coordinates of (0.18, 0.27) were also achieved in the host-free Au(I)-1-based OLED.

As depicted in Figure 9, by modifying **a5,** whose HOMO and LUMO are localized on the aryl donor triphenylamine (TPA) and mainly on the C^N^C ligand [73], respectively, two blue-emitting Au(III)-TADF complexes were developed by lowering the HOMO energy or raising the LUMO energy. Specifically, Au(III)-1 was prepared by replacing the OEt group in complex **a5** with a more electron-donating dimethyl-amine group to raise the LUMO energy, while Au(III)-2 was prepared by further substituting the phenyl rings of the TPA with electron-withdrawing F substituents to lower the HOMO energy. By means of electrochemistry, the LUMO energy of Au(III)-1 and Au(III)-2 was measured to be 0.18 eV higher than that of complex **a5**, attributable to the replacement of the OEt group by the more electron-donating NMe2 group. For Au(III)-2, the fluorine substituents on the TPA ligand slightly lowered the HOMO energy by about 0.03 eV from that of Au(III)-1. Thus, PL maxima at 484 and 470 nm were achieved in Au(III)-1 and Au(III)-2, respectively, in PMMA thin film. Despite its less blue emission, the high PLQY of 0.82 for Au(III)-1 in PMMA thin film is much higher than that (0.34) of Au(III)-2. By their application in solution-processed OLEDs, EL properties of both TADF Au(III) complexes were investigated. The device structure was ITO/PEDOT:PSS/OTPD (4 nm)/PYD2: Au-emitter (60 nm)/DPEPO (10 nm)/TPBi (40 nm)/LiF (1.2 nm)/Al (100 nm). The Au(III)-1- and Au(III)-2-based devices show EL maxima at 473 and 465 nm, respectively, with a FWHM of 64–67 nm and CIE coordinates of (0.16, 0.25) for the former and (0.16, 0.23) for the latter. Due to its higher PLQY, the EQE_max_ of 15.25% shown by the Au(III)-1-based device is much higher than that of the Au(III)-2-based device (6.76%). By employing an MR-TADF blue emitter, v-DABNA, as the terminal emitter, a hyper-fluorescent solution-processed OLED was fabricated using the same device structure for Au(III)-1 (Figure 10). Efficient ET from Au(III)-1 to v-DABNA, with an estimated Förster distance of 2.83 nm and an ET rate of 2.35 × 10^7^ s^−1^, was determined by photophysical investigations. The blue hyper-fluorescent OLED achieved a high EQE_max_ of 16.6%, a narrowband emission with an FWHM of 23 nm, and high color purity with CIE coordinates of (0.14, 0.18). When compared with that (30%) of the Au(III)-1 only device, the efficiency roll-off (13%) of the hyper-fluorescent device was significantly improved, attributable to the effective ET process in the latter that could lower the population of triplet excitons in Au(III)-1 at high luminance [40].

Since the reports of CMA emitters based on bulky NHC ligands, with emission quantum yields up to unity, a k_r_ of up to 1 × 10^6^ s^−1^, and EL efficiency up to 20%, Cu(I) complexes have been expected to replace expensive phosphorescent complexes in OLEDs as emitters [64,67,70,74]. Nonetheless, the operational stability of Cu(I)-based OLEDs, especially that of blue-emitting ones, has barely been reported in the literature [75]. With the use of heterocyclic π-annulated NHC ligands with appreciable π-acidity, we report a panel of thermally and air-stable CMA(Cu)-TADF emitters based on bulky pyrazine- and pyridine-fused NHC ligands [62]. As a blue-emitter, Cu-5 (Figure 11) shows PL emission with a maximum at 470 nm, a PLQY of 0.52, and a k_r_ of 1.1 × 10^6^ s^−1^ in thin films. EL performance of Cu-5 was investigated in a vacuum-deposited OLED with a device structure of ITO/HAT-CN (5 nm)/TAPC (50 nm)/TCTA (10 nm)/Cu-5: TCTA: DPEPO (20 nm)/DPEPO (10 nm)/TPBi (40 nm)/LiF (1 nm)/Al (100 nm). When the doping concentration was 4 wt%, the Cu5-based device displayed a blue emission with an EL maximum located at 474 nm and CIE coordinates of (0.14, 0.22), as well as a high EQE_max_ of 23.6%, which is among the best blue OLEDs reported in the literature [76,77]. Notably, the operational lifetime (LT_90_) of OLEDs based on Cu5 was 0.4 h at an L_0_ of 7600 cd m^−2^, corresponding to 11.6 h at an L_0_ of 1000 cd m^−2^ using the acceleration coefficient (n), obtained as shown in Figure 11. By using v-DABNA as a terminal emitter, hyper-fluorescent OLEDs with Cu5 as a sensitizer showed an improved color purity with CIE coordinates of (0.15, 0.20) and a longer LT_90_ of 12.2 h at an L_0_ of 1000 cd m^−2^.

## 5. Outlook and Perspective

Bimolecular annihilations, such as TTA and TPA, caused by the long decay time of TADF emitters are the main reason for the efficiency roll-off and unsatisfactory stability of TADF-OLEDs, particularly blue ones. To surmount such disadvantages, specific molecular structures with extremely small or even negative Δ*E*_ST_ have been designed to decrease the decay time [78,79,80,81,82].

Yersin and co-workers reported a strategy to design a new type of emitter based on a new exciton harvesting mechanism by significantly reducing Δ*E*_ST_, rigidifying molecular structure, maximizing the Franck–Condon factors, shifting the charge transfer states, and providing energetically nearby lying states [78]. As shown in Figure 12, in the designed compound DSH, both D and A moieties are rigidly linked by two bridges at relatively large separations [81]. Thus, the low-lying CT states, ^1^CT(DA) and ^3^CT(DA), have a minimal splitting of ≈10 cm^−1^ (1.2 meV), while the fluorene-substituted bridge can further stabilize the molecular structure. At the same time, one more low-lying fluorene-localized ^3^LE(F) state is found in the fluorene-substituted bridge. Their design mitigated the forbidden ISC processes, enabling faster ISC and RISC rates for DSH than those of conventional TADF emitters. Thus, the long decay times of conventional TADF emitters can be avoided in DSH. By using DSH as an emitting dopant in OLEDs, with a structure of ITO/HAT-CN (5 nm)/TAPC (40 nm)/CCP (10 nm)/DSH: PPF (10 nm)/PPF (10 nm)/Tm3PyBPZ (40 nm)/LiF (1.2 nm)/Al (100 nm), an EQE_max_ of 18.7% with an EL maximum at 486 nm was achieved. Although the device performance and emission color are strongly dependent on the dielectric constant of the host material, which could cause difficulty in optimizing the device structure, the specific molecular design of DSH provides a guideline for next-generation TADF materials.

Despite the general agreement that Δ*E*_ST_ must be positive, some nitrogen-substituted phenalene analogues have the potential for a negative Δ*E*_ST_, which can be attributed to the double-excitation configurations. Two electrons of occupied orbitals are promoted to virtual orbitals in double-excitation configurations [83,84,85,86]. Since the accessible double-excitation configurations in T_1_ are restricted (Pauli exclusion principle), an effective admixture of such configurations stabilizes S_1_, leading to a possibly negative Δ*E*_ST_ when the stabilization overcomes the exchange energy (Figure 13a). With the motivation of discovering molecules with negative ΔE_ST_, from which delayed fluorescence from inverted singlet and triplet (DFIST) excited states can be observed, Miyajima and colleagues computationally screened 34,596 molecules, which were heptazine analogues with 186 different substituents [82]. Among the screening results, HzTFEX_2_ and HzPipX_2_ (Figure 13b), were compared. Theory (EOM-CCSD) calculations revealed that HzTFEX_2_ has a negative Δ*E*_ST_ of −12 meV, while HzPipX_2_ has a small positive Δ*E*_ST_ of 10 meV. Measurements of the photophysics and transient PL decay further confirmed the negative ΔE_ST_ of HzTFEX_2_, which exhibits DFIST. When it was used as the emitting dopant in the device with a structure of Glass/ITO (130 nm)/PEDOT: PSS (30 nm)/MoO_3_ (5 nm)/BST (3 nm)/DBFSiDBF (10 nm)/PPF: 10 wt% HzTFEX_2_ (15 nm)/PPF (10 nm)/Alq3 (40 nm)/Liq (1 nm)/Al (80 nm), intense blue EL originating from HzTFEX2 was observed with an EQE_max_ of 17% and EL maxima at 450 nm and 479 nm with CIE coordinates of (0.17, 0.24). The remarkable efficiency roll-off suggests that great efforts are still needed to fully exploit the benefits of negative ΔE_ST_ in EL devices.

## Figures and Tables

**Figure 1 micromachines-13-02150-f001:**
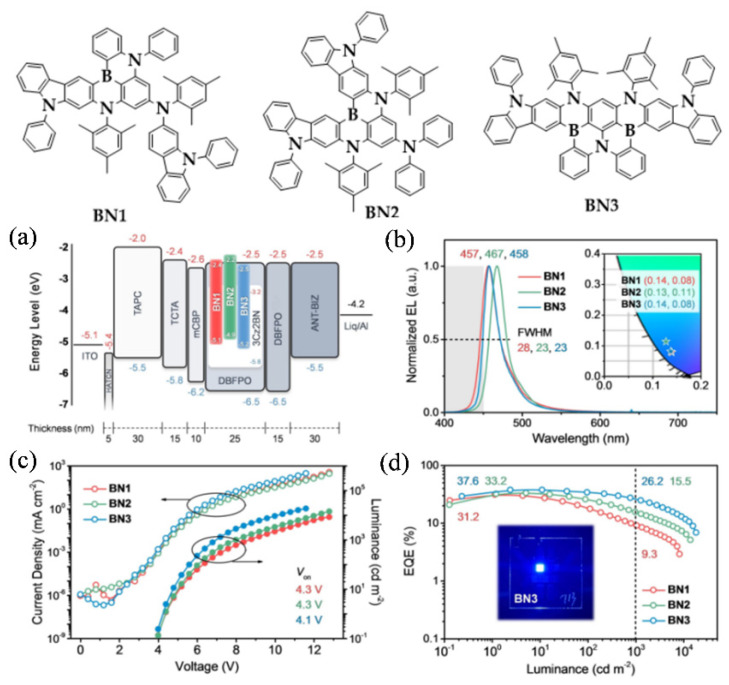
Molecular structures of BN1–BN3. (**a**) Device architecture for MR-TADF OLEDs with energy level alignment of the relevant material, (**b**) EL spectra, with CIE coordinates shown in the inset, (**c**) J–V–L characteristics, and (**d**) EQE–luminance curves of MR-TADF OLEDs. Reproduced with permission from Xialei Lv, Jingsheng Miao, Meihui Liu et al., “Extending the π-Skeleton of Multi-Resonance TADF Materials towards High-Efficiency Narrowband Deep-Blue Emission”; published by *John Wiley and Sons*, **2022**. [45].

**Figure 2 micromachines-13-02150-f002:**
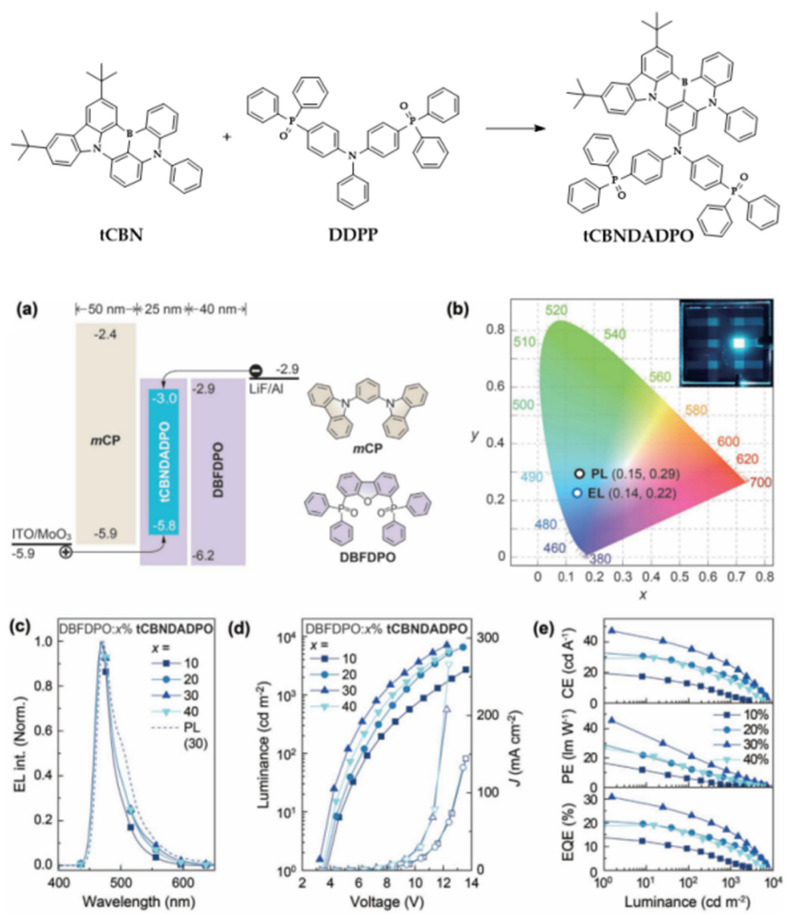
Host–guest integration strategy for constructing tCBNDADPO to combine high color purity and efficiency. (**a**) Energy level diagram of devices and chemical structures of host and carrier transporting materials; (**b**) CIE coordinates of PL and EL emissions from DBFDPO:30% tCBNDADPO and device photo at 5 V; (**c**) concentration dependence of EL spectra and comparison to PL spectrum of 30% doped film; (**d**) variation of current density (J, hollow symbols)–voltage–luminance (solid symbols) characteristics at different x%; (**e**) dependence of efficiency versus luminance curves on x%. Reproduced with permission from Jinkun Bian, Su Chen, Lili Qiu et al., “Ambipolar Self-Host Functionalization Accelerates Blue Multi-Resonance Thermally Activated Delayed Fluorescence with Internal Quantum Efficiency of 100%”; published by *John Wiley and Sons*, **2022** [44].

**Figure 3 micromachines-13-02150-f003:**
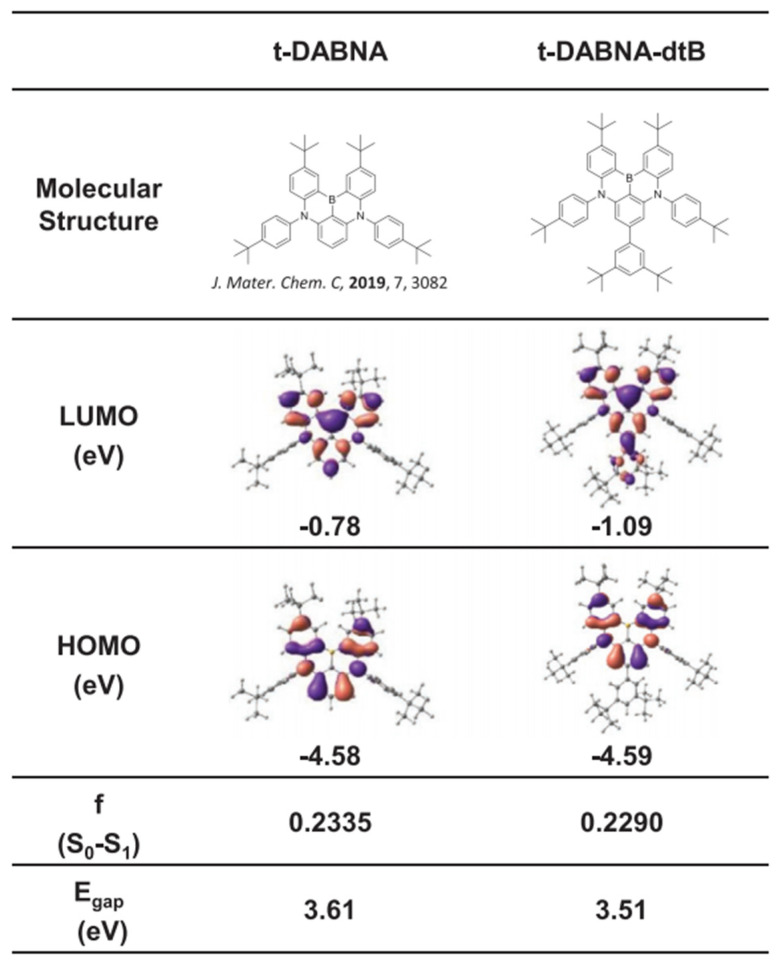
Molecular structures and DFT calculations of t-DABNA and t-DABNA-dtB. Reproduced with permission from Jinho Park, Ki Ju Kim, Junseop Lim et al., “High Efficiency of over 25% and Long Device Lifetime of over 500 h at 1000 nit in Blue Fluorescent Organic Light-Emitting Diodes”; published by *John Wiley and Sons*, **2022** [49].

**Figure 4 micromachines-13-02150-f004:**
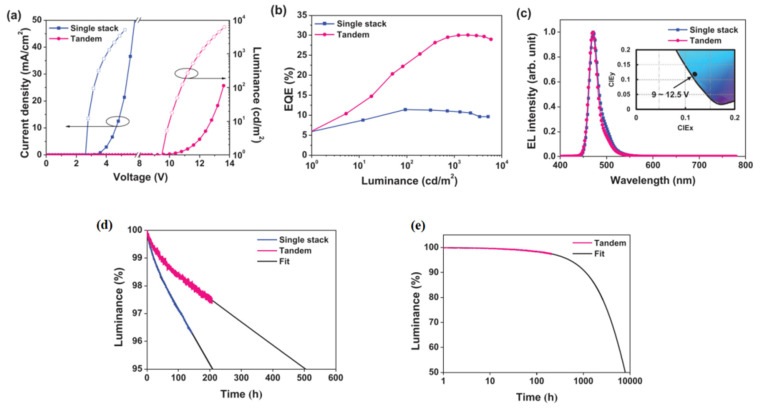
(**a**) J–V–L and (**b**) EQE-L characteristics, (**c**) EL spectra and CIE coordinates (inset), and (**d**) lifetime data of single stack and tandem devices. (**e**) Extrapolated plot of the tandem device. Reproduced with permission from Jinho Park, Ki Ju Kim, Junseop Lim et al., “High Efficiency of over 25% and Long Device Lifetime of over 500 h at 1000 nit in Blue Fluorescent Organic Light-Emitting Diodes”; published by *John Wiley and Sons*, **2022** [49].

**Figure 5 micromachines-13-02150-f005:**
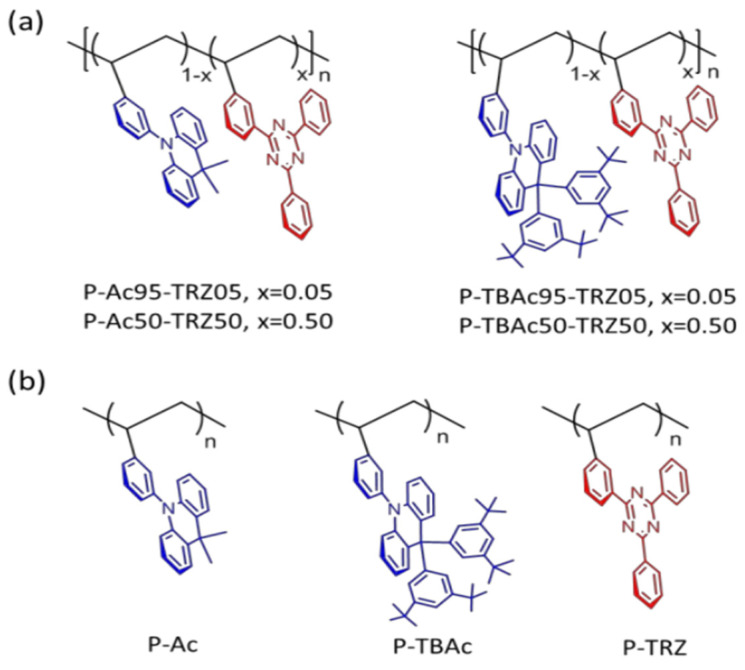
Molecular structures of (**a**) TSCT-based polymers and (**b**) the corresponding control polymers. Reprinted with permission from Shao, S.; Hu, J.; Wang, X.; Wang, L.; Jing, X.; Wang, F. “Blue Thermally Activated Delayed Fluorescence Polymers with Nonconjugated Backbone and Through-Space Charge Transfer Effect” [J]. *J. Am. Chem. Soc*. **2017**, 139, 17739−17742. Copyright 2017 American Chemical Society [59].

**Figure 6 micromachines-13-02150-f006:**
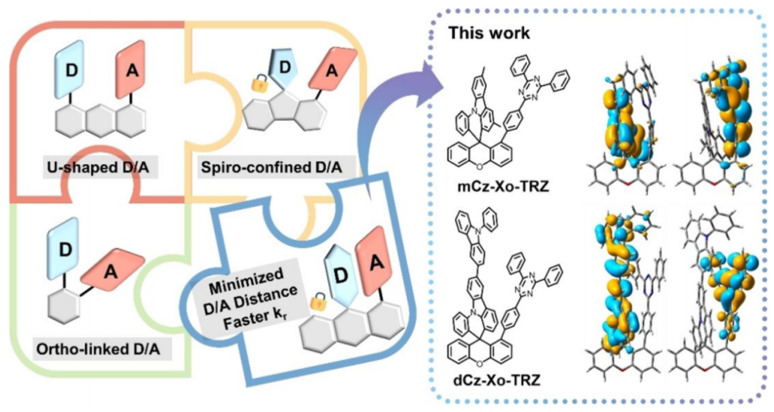
TSCT-TADF molecules with different types of donor–acceptor alignments, and molecular structures of mCz-Xo-TRZ and dCz-Xo-TRZ with their HOMO/LUMO distribution. Reproduced with permission from Tianyu Huang, Qi Wang, Guoyun Meng et al., “Accelerating Radiative Decay in Blue Through-Space Charge Transfer Emitters by Minimizing the Face-to-Face Donor–Acceptor Distances”; published by *John Wiley and Sons*, **2022** [39].

**Figure 7 micromachines-13-02150-f007:**
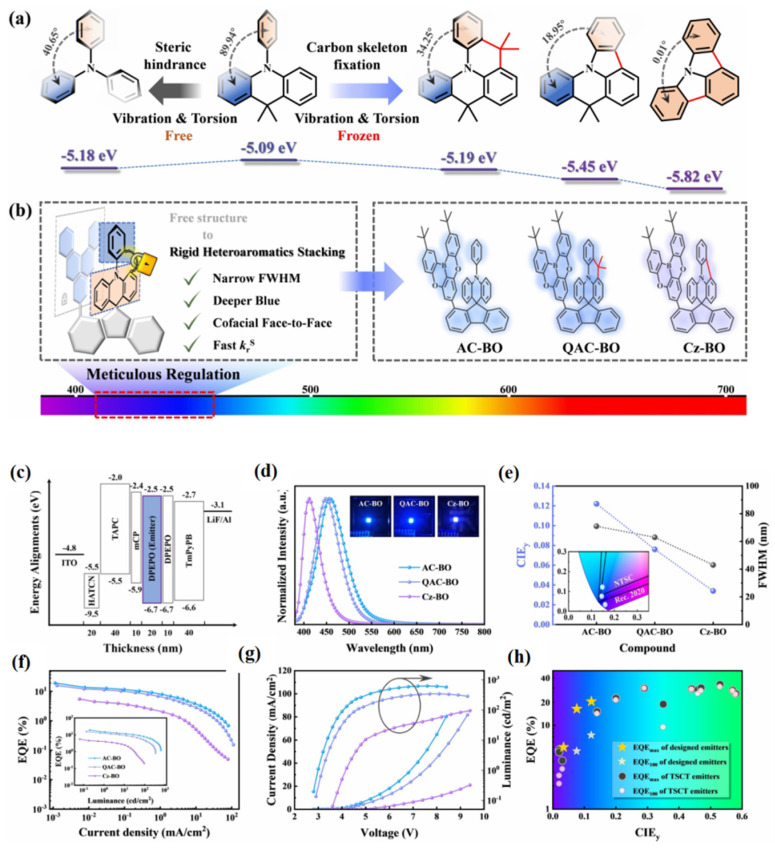
(**a**) The relationship between structures and HOMO levels of phenyl acridine and its skeleton-modified derivatives based on the DFT simulations. (**b**) Design strategy for controlling intramolecular stacking of rigid heteroaromatic compounds and the chemical structures of the designed compounds. (**c**) Device configuration and the related energy levels. (**d**) EL spectra with the photographs of devices (inset). (**e**) CIE_y_ and FWHM of the EL spectra with CIE plot of EL spectra (inset). (**f**) EQE–current density curves with EQE–luminance curves (inset). (**g**) Current density–voltage–luminance curves. (**h**) EQE_max_s and EQEs at 100 cd m^−2^ (EQE_100_) of this work and the reported TSCT emitters. Reproduced with permission from Zhennan Zhao, Cheng Zeng, Xiaomei Peng et al., “Tuning Intramolecular Stacking of Rigid Heteroaromatic Compounds for High-Efficiency Deep-Blue Through-Space Charge-Transfer Emission”; published by *John Wiley and Sons*, **2022** [60].

**Figure 8 micromachines-13-02150-f008:**
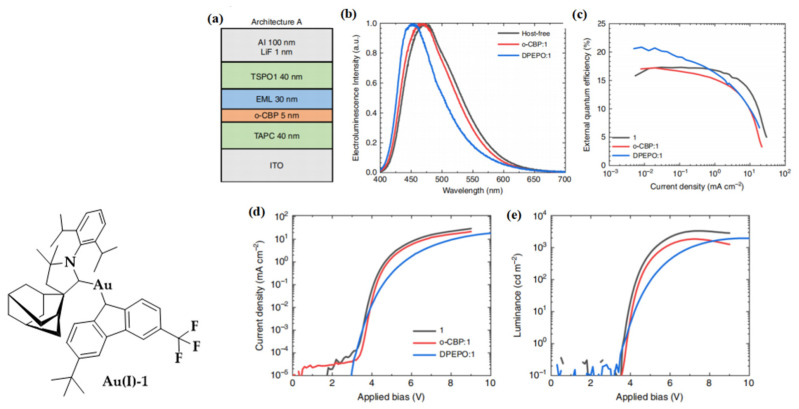
Molecular structure of Au(I)-1. (**a**) Schematic OLED architecture used; (**b**) normalized EL spectra, (**c**) EQE–current density, (**d**) current–voltage, and (**e**) luminance–voltage characteristics of OLEDs with Au(I)-1 in host-free and host-guest structures. Reproduced with permission from Patrick J. Conaghan et al., “Highly efficient blue organic light-emitting diodes based on carbene-metal-amides”; published by *Springer Nature*, **2020** [71].

**Figure 9 micromachines-13-02150-f009:**
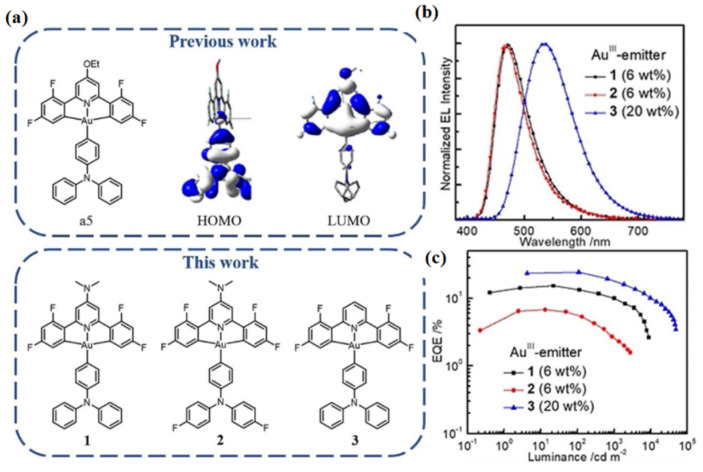
(**a**) Top: previously reported arylgold(III)-TADF complex **a5** and its HOMO and LUMO surfaces. Bottom: donor–acceptor type C^N^C gold(III) complexes Au(III)-1–Au(III)-3 in this work; (**b**) normalized EL spectra; and (**c**) EQE–luminance characteristics of OLEDs based on Au(III)-1, Au(III)-2, and Au(III)-3 at their optimized doping concentration. Reproduced with permission from Dongling Zhou, Gang Cheng, Glenna So Ming Tong et al., “High Efficiency Sky-Blue Gold(III)-TADF Emitters”; published by *John Wiley and Sons*, **2020** [72].

**Figure 10 micromachines-13-02150-f010:**
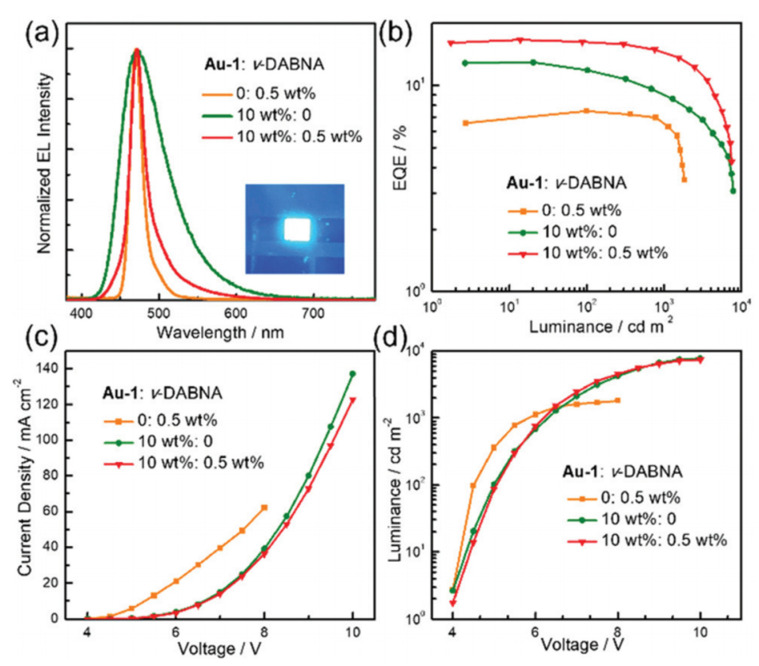
(**a**) Normalized EL spectra, (**b**) EQE–luminance, (**c**) current density–voltage, and (**d**) luminance–voltage characteristics of OLEDs with different concentrations of Au(III)-1 and v-DABNA. The image of the sensitized OLED is shown in the inset of (**a**). Reproduced from Ref. [40] with permission from the Royal Society of Chemistry.

**Figure 11 micromachines-13-02150-f011:**
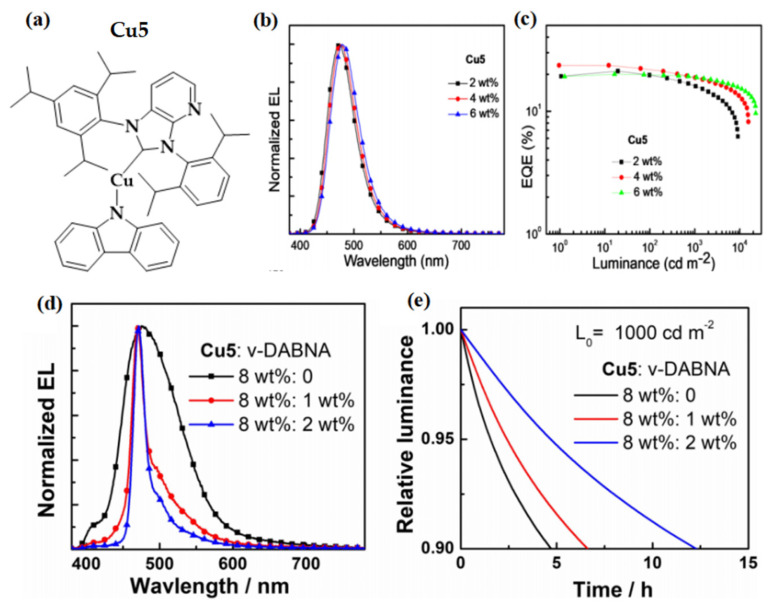
(**a**) Molecular structure of Cu5; (**b**) normalized EL spectra; (**c**) EQE–luminance of OLEDs based on Cu5 with various doping concentrations ranging from 2 to 6 wt%; (**d**) EL spectra; and (**e**) relative luminance versus operation time at an L_0_ of 1000 cd m^−2^ of the hyper-OLEDs with Cu5 and v-DABNA. Reproduced with permission from Rui Tang, Shuo Xu, Tsz-Lung Lam et al., “Highly Robust CuI-TADF Emitters for Vacuum-Deposited OLEDs with Luminance up to 222 200 cd m^−2^ and Device Lifetimes (LT90) up to 1300 hours at an Initial Luminance of 1000 cd m^−2^”; published by *John Wiley and Sons*, **2022** [62].

**Figure 12 micromachines-13-02150-f012:**
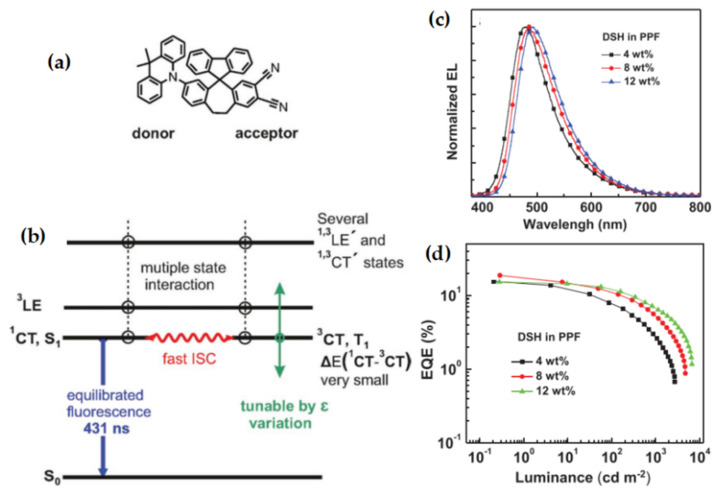
(**a**) Molecular structure of DSH; (**b**) simplified energy level diagram for the DSH molecule; (**c**) normalized EL spectra and (**d**) EQE–luminance characteristics of OLEDs with different concentrations of DSH. Reproduced with permission from Hartmut Yersin, Rafał Czerwieniec, Larisa Mataranga-Popa et al., “Eliminating the Reverse ISC Bottleneck of TADF Through Excited State Engineering and Environment-Tuning Toward State Resonance Leading to Mono-Exponential Sub-µs Decay. High OLED External Quantum Efficiency Confirms Efficient Exciton Harvesting”; published by *John Wiley and Sons*, **2022** [81].

**Figure 13 micromachines-13-02150-f013:**
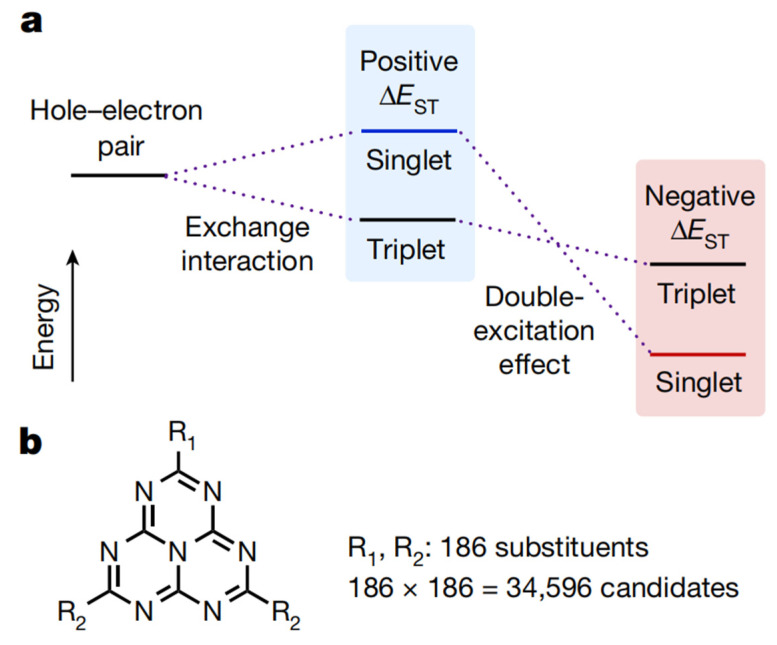
(**a**) Schematic diagram of singlet and triplet excited states split in energy by the exchange interaction (middle) and then inverted by including the double-excitation effect (right); (**b**) molecular structures of the heptazine analogues examined in the computational screening. Reproduced with permission from Naoya Aizawa et al., “Delayed fluorescence from inverted singlet and triplet excited states”; published by *Springer Nature*, **2022** [82].

## Data Availability

Not applicable.

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
