# Peer review of "Recent Progress in Blue Thermally Activated Delayed Fluorescence Emitters and Their Applications in OLEDs: Beyond Pure Organic Molecules with Twist D-π-A Structures"

_micromachines, 2022, doi:10.3390/mi13122150_

Round 1

Reviewer 1 Report

In this review article, the authors report the recent progress of blue thermally activated delayed fluorescent emitters and their applications in OLEDs. The nature of the review is significant and interests the readers of Micromachines. I believe that the document can be published after minor revisions. Several issues must be solved before it can be considered for publication.

1.     The text needs significant editing to correct the numerous grammar and typo errors found throughout the document.

2.     The quality of the figures needs to be improved.

3.     The authors have added many unnecessary figures for reference purposes only. They should describe in the running text in much more detail, or Inappropriate figures should be removed.

4.     Page 1: In the abstract, acronyms are not required as they don’t appear again in the abstract. In addition, they are again described in the main script.

5.     Page 1, Line 38-44: Authors mention the device architecture is for phosphorescent and TADF OLEDs. However, similar device architectures are also used for other OLEDs, say fluorescent. In view of the readers, it should be changed accordingly.

6.     Page 1-3: In the Introduction, the authors fail to explain the need for blue OLEDs and the issues related to blue OLEDs in fluorescent and phosphorescent OLEDs. And as to how TADF OLEDs are beneficial in solving the issue. As this review concerns Blue OLEDs, I strongly feel it will benefit the readers.

7.     Page 9: In view of the readers, the phenomena in Figure 6 should be explained in detail.

Reviewer 2 Report

The manuscript “Recent progress of blue thermally activated delayed fluores-2 cence emitters and their applications in OLEDs: beyond the 3 pure organic molecules with twist D-π-A structures” demonstrated the development and future challenges for OLEDs based on thermally activated delayed fluorescence (TADF) materials. The manuscript is meaningful for OLEDs’ researches. After minor revision, it may be considered acceptable by Micromachines.

1.        The legend format needs to be consistent. The format of Figure 2 is different from Figure 1.

2.        For these OLEDs, what are the limitation to the lifetime? The lifetime was an important factor for LEDs. Does the TADF materials affect the stability of OLEDs. Please add some introduction.

3.        There are some grammars relating issues in the manuscript. The authors should solve those issues in the revision.

4.        I suggest the authors to add a comparison table and compare the previously reported OLEDs based on blue thermally activated delayed fluorescence in the table.

Reviewer 3 Report

Thermally activated delated fluorescence (TADF) materials is a key element of OLEDs technology in terms of their efficiency, lifetime, and sustainability. In this context, the literature by the author will be attracted through many readers as the paper have summarized the recent trends of TADF very effectively. The paper is acceptable for the journal after few recommendations is presented as below. 

1.     In line 40-44, the roles of auxiliary layers should be explained separately for the readers’ understanding. (Recommendation)

2.     Author should Insert a table to list up the efficiency of the OLEDs in this paper and recent research results.  
